# A Descriptive Review of the Impact of Patient Motion in Early Childhood Resting-State Functional Magnetic Resonance Imaging

**DOI:** 10.3390/diagnostics12051032

**Published:** 2022-04-20

**Authors:** Jenna Schabdach, Rafael Ceschin, Vanessa Schmithorst, M. Dylan Tisdall, Aaron Alexander-Bloch, Ashok Panigrahy

**Affiliations:** 1Children’s Hospital of Philadelphia, Philadelphia, PA 19104, USA; aaron.alexander-bloch@pennmedicine.upenn.edu; 2School of Medicine, University of Pittsburgh, Pittsburgh, PA 15213, USA; rafael.ceschin@pitt.edu; 3UPMC Children’s Hospital of Pittsburgh, Pittsburgh, PA 15224, USA; vanessa.schmithorst@chp.edu (V.S.); ashok.panigrahy@chp.edu (A.P.); 4Perelman School of Medicine, University of Pennsylvania, Philadelphia, PA 19104, USA; mtisdall@pennmedicine.upenn.edu

**Keywords:** rs-fMRI, pediatric rs-fMRI, patient motion, task-free fMRI

## Abstract

Resting-state functional magnetic images (rs-fMRIs) can be used to map and delineate the brain activity occurring while the patient is in a task-free state. These resting-state activity networks can be informative when diagnosing various neurodevelopmental diseases, but only if the images are high quality. The quality of an rs-fMRI rapidly degrades when the patient moves during the scan. Herein, we describe how patient motion impacts an rs-fMRI on multiple levels. We begin with how the electromagnetic field and pulses of an MR scanner interact with a patient’s physiology, how movement affects the net signal acquired by the scanner, and how motion can be quantified from rs-fMRI. We then present methods for preventing motion through educational and behavioral interventions appropriate for different age groups, techniques for prospectively monitoring and correcting motion during the acquisition process, and pipelines for mitigating the effects of motion in existing scans.

## 1. Introduction

Functional magnetic resonance images (fMRIs) are used to map neuronal networks in a patient’s brain either at rest or during a specific task. The operation of neuronal networks is observed by measuring changes in deoxygenated blood flow throughout the brain: brain areas with higher concentrations of deoxygenated blood are active in response to the behavior of the patient at that moment. These behaviors can be responses to tasks and stimuli often used during fMRI scans, which can include listening to a sound, looking at a light, calculating arithmetic operations, and performing memory exercises. However, the human brain is active even when the person is not performing a directed task. fMRIs performed while the patient is in a task-free state are called resting-state fMRIs (rs-fMRIs). rs-fMRIs record the activity which occurs in the underlying networks of a patient’s brain. While they are not commonly used in clinical applications, they have been shown to have applications to provide information about a patient’s neurodevelopmental outcomes. In particular, rs-fMRIs have been proposed to have potential for aiding in the diagnosis and/or for characterizing the neural underpinnings of autism and attention deficit hyperactivity disorder as well as for improving our understanding of the clinical pathophysiology of disorders such as Alzheimer’s disease, epilepsy, and multiple sclerosis [1,2,3].

The purpose of fMRI scans is to observe brain activity via the global and local firing of neurons. The firing of neurons leads to increased blood flow (neurovascular coupling) which is strongly locally controlled in response to the concentration of oxygen and carbon dioxide in that area of brain tissue. When a specific region of brain tissues increases its activity, the initial usage of oxygen causes a local decrease in oxygenated hemoglobin, an increase in deoxygenated hemoglobin, and an increase in carbon dioxide. After a slight delay of a few seconds, cerebral blood flow increases to deliver a supply of oxygenated hemoglobin and remove deoxygenated hemoglobin. This large rebound in local tissue oxygenation is highly detectable in fMRIs and is referred to as the blood oxygen level-dependent (BOLD) signal. Though precisely recording the activity of individual neurons is beyond the spatial and temporal resolution of existing MRI technology, the BOLD signal can serve as an approximation of the patient’s brain activity.

One of the biggest challenges with rs-fMRIs is their sensitivity to patient motion. Herein, we present a review of rs-fMRIs, patient motion, and research surrounding patient motion in rs-fMRIs. This work is intended as a descriptive overview of the range of issues related to motion in rs-fMRIs in a comprehensive, scientifically sound, and approachable document intended for anyone in the field of fMRI spanning not only clinicians and neuroimaging researchers but also students, radiology technicians, physicists, and engineers. Specifically, we describe the contents of rs-fMRIs from physiological, biophysical, and computational perspectives, outline three effects of patient motion on rs-fMRI sequences, and detail the evolution of methods used to prevent and correct motion in pediatric MR imaging. This review is not meant to be systematic. We begin with a discussion of the biological signals recorded in an rs-fMRI, the physiological and electromagnetic (EM) factors which affect these signals, and the data structure used to digitally store these signals. We then discuss the impact of motion on the recorded signals. The majority of this paper focuses on methods that have been developed to prevent and compensate for patient motion in rs-fMRIs.

## 2. Brain Activity and Resting-State Functional Magnetic Resonance Imaging

In vivo studies of brain activity involve observing the changes in the activations present in a patient’s brain related to a task or stimuli compared to the activations present when the patient is in a task-free state. Extensive observation of task-free brain activity in early fMRI studies led researchers to theorize and later confirm the existence of a default mode network of neurons that is active during task-free states [4,5,6]. To further study the default mode network, researchers turned to rs-fMRI. Patterns in the fluctuations of the BOLD signals recorded during rs-fMRIs were used to map the default mode network.

The concept of the healthy default mode network and its measurement using rs-fMRI has expanded the horizons of neurocognitive research. Discoveries made using rs-fMRIs have deepened our understanding of traumatic brain injuries as injuries to neuronal networks, of the mechanisms of neurodegeneration occurring in Alzheimers, and of the heterogeneity of autism spectrum disorder [7,8,9,10,11]. As the scientific community learns more about functional connectivity in the brain, we are beginning to identify gaps both in our knowledge and in the data needed to acquire more knowledge. Specifically, we need large quantities of high quality rs-fMRIs across the human lifespan. One of the greatest barriers to achieving this data goal is the sensitivity of rs-fMRIs to patient motion. We devote the remainder of this section to a discussion of the computational form of rs-fMRIs and how the electrophysical properties of tissue are translated by the scanner into a visualizable form.

On a computational level, an rs-fMR image is a four-dimensional image sequence. The first three dimensions of the image are spatial (i.e., head-foot, left-right, anterior-posterior) while the fourth dimension is time. The spatial dimensions are represented as volume elements, which are three-dimensional versions of pixels known as voxels. The temporal signal can be more difficult to conceptualize. The sequence can be thought of as an ordered list of 3D image volumes where each voxel contains a decimal value or as a single 3D image volume where each voxel contains a temporal signal rather than a single decimal value. An example of these parallel views can be seen in Figure 1. One aspect of rs-fMRIs illustrated in this figure is that an rs-fMRI is a discrete representation of a continuous signal. Generally, the spatial resolution of an rs-fMRI is on the order of millimeters and the temporal resolution is on the order of seconds. These resolutions could make it simple to assume that the intensity of a given voxel is uniform across the cubic millimeter for the complete two to three seconds required to record the signal intensities in all voxels of the image volume. This false assumption may simplify some conceptualization around rs-fMRIs but is not accurate: the space designated as belonging to a single voxel may be composed of a range of signals, from a large quantity of neurons (which are generally between 4 and 100 microns in diameter, excluding axons), and potentially from different tissue types. The combination of signals is recorded as a single value at a single time point as part of the conversion from the continuous domain to the discrete domain.

The signals recorded during an rs-fMRI are based on the magnetic properties of the molecules in and around the object being scanned. Molecules with low proton density, such as air, appear darker in the rs-fMRI than molecules with greater proton density. In rs-fMRIs of the brain, the air outside the patient’s head appears dark while soft tissues such as fat appear much brighter. The appearances of these tissues serve as the base MR signal in BOLD-weighted scans while brain activity is associated with variations in the BOLD-weighted signal.

Kim et al. performed an in vivo comparison of BOLD fMRI signal and neuronal activity in animal models. They found that BOLD signal has an approximately linear correlation with single-unit neuronal activity when the BOLD signal is sampled from a large enough region [12]. This relationship can be seen in spatial resolutions on the order of millimeters, but it deteriorates at more granular physical resolutions due to the biophysical properties of neurons. Notably, the authors also report that the degree of the correlation varies depending on voxel location.

BOLD signal is not the only physiological signal recorded during an rs-fMRI scan. Cardiac and respiratory activity contribute to the resulting image. Cardiac activity involves changes in oxygen concentration, deformation and pulsation of blood vessels, and movement of cerebrospinal fluid [13]. Shmueli et al. showed that the noise due to cardiac activity is about 1% of the variation in a BOLD-weighted sequence, which is similar in magnitude to the BOLD signal [14]. The impact of cardiac noise varies depending on the location of each voxel in the brain [15]. Respiratory activity contributes to these effects, but also causes slight changes in the patient’s position.

To add to uncertainty about the true location of brain activity-related BOLD signal, individual MR scanners have their own unique patterns of distortions which impact the effective signal. These distortions are recorded in a process called B0 field mapping. The B0 field map is essentially a 3D plot of the magnetic field inside the bore of the scanner. It can be used to perform techniques such as shimming, which are designed to reduce the impact of B0 field inhomogeneities inherent to the scanner.

During analysis of an rs-fMRI, separation of the BOLD signal from the background signals and confound signals is performed. The filtered signal is used to identify regions of the brain which are active at the same time. Certain groups of known regions have been correlated with healthy, baseline brain activity. As the patient was not performing a task with a specific stimulus during the scan acquisition, these active areas are indicative of resting-state networks.

## 3. Three Effects of Patient Motion on rs-fMRIs

A patient may move during an rs-fMRI scan for a variety of reasons. An rs-fMRI is often performed during the same session as other MRI scan types. MRI scanning sessions can be quite lengthy. It follows that two reasons a patient may move during this time are physical discomfort or boredom. Younger patients in particular are highly susceptible to growing restless during an rs-fMRI scan. Feelings of distress due to claustrophobia, stress, surprise, or fear may contribute to the patient’s conscious and/or unconscious desire to move. Even if the patient is completely comfortable and calm during the scan, small movements will persist due to regular bodily functions such as cardiac activity and respiration.

Regardless of the reason, patient movement affects three key aspects of the acquired rs-fMRI scan. The three effects can be categorized as the positional effect, the spin history effect, and the susceptibility effect.

### 3.1. Positional Effect

The first effect of patient motion is the positional effect of motion. The most important assumption when analyzing rs-fMRIs is that each voxel recorded a signal from the same locations in the brain throughout the entire scan. This assumption is immediately violated the moment the patient moves, confounding the analysis. Even the smallest movement causes at least some of the voxels in the rs-fMRI to record signals from a different location in the patient’s brain. It may seem that the positional impact of motion should scale with respect to the magnitude of the motion, but this is not the case: even half a millimeter of movement is enough to cause the signal in a voxel to arise from adjacent tissues or brain regions rather than the space they were originally recording.

### 3.2. Spin History Effect

The spin history effect is related to the interactions between protons and the magnetic field during an MR scan. When the patient first enters the MR scanner, the net magnetic moment of the protons in the body produce a small net magnetic moment aligned along the scanner’s main magnetic field (i.e., down the scanner bore). Then, per protocol instructions, bands of protons have their net magnetic moments tipped away from the main field by a radiofrequency (RF) “excitation” pulse, causing them to rotate, or “precess”, around the main magnetic field. When the RF pulse ends, the tipped protons relax back to their alignment with the primary magnetic field at rates dependent on the properties of their microscopic environments (e.g., grey matter and white matter exhibit different relaxation times). During this period of relaxation, the precession of the protons’ net magnetic moments induces an electric signal in the receive coils of the MR scanner, which is recorded as the signal produced by the tissues in the activated band and used to generate the resulting image.

The time between RF pulses is planned under the assumption that the patient will remain still during the scan. When the patient moves, the protons in the activated band will also move. The following process is illustrated in Figure 2. During the next RF pulse, any excited protons that moved from the previously activated band into the subsequent band will be further excited. The signals the twice-excited protons produce during the relaxation period will be much darker than they should be due to incomplete relaxation between the pulses, resulting in band-shaped shadows. Similarly, slices that miss an excitation pulse due to movement will be brighter, due to more-complete relaxation between pulses.

The phrase “spin history effect” refers to the false variations in the recorded signal due to the incorrect timing of pulses that the protons experienced. The spin history effects have a short-term impact on the image sequence: they can affect the signal in volumes acquired up to 10 s after the patient moves. During that period, the proton’s net magnetic moments recover to the state appropriate for their new locations [16].

### 3.3. Susceptibility Effect

The susceptibility of a material describes how the materials respond to a magnetic field. Most materials have some degree of paramagnetic or diamagnetic properties. The dipoles in paramagnetic materials align with the magnetic field while the dipoles in diamagnetic materials anti-align with the magnetic field. Both materials cause dipole fields which change the overall magnetic field everywhere. These distortions are most prominent in strong magnetic fields at locations where two materials with different susceptibilities interface.

When the patient remains stationary, the magnetic distortions due to susceptibility remain constant and do not contribute to changes in the BOLD signal. Any recorded signal related to susceptibility essentially functions as a constant offset in the image sequence. However, when the patient moves, the locations of materials with different susceptibilities also move. As a result, the signal offsets provided by the susceptibility properties in each voxel also change. The changes in the susceptibility-related signal recorded by the scanner are of a magnitude which can lead to spurious correlations during the analysis of the image sequence.

In contrast to spin history effects, the susceptibility effects of motion are present throughout the entire image sequence. The only way to change the susceptibility-related signal in any specific voxel is to change the area of the brain from which the voxel is recording electromagnetic signals.

### 3.4. Summary

Most early work in addressing the effects of patient motion in rs-fMRIs focuses on the positional effects of motion. The impact of the spin history effect and the susceptibility effect, loosely categorized as the signal effects of motion, has been increasingly addressed by work over the last decade. In the remainder of this work, we will focus on techniques developed to address and mitigate these effects of patient motion.

## 4. Measuring Patient Motion

Thus far, we have discussed rs-fMRIs and the effects of patient motion on them, but we have yet to define how motion is measured in an rs-fMRI. Two types of measurements are often jointly used by the rs-fMRI community. These measurements are used to quantify (1) the overall change in the patient’s position and (2) the overall change in the recorded signal.

Several different researchers have proposed different methods for calculating the change in the patient’s position, though they follow the same general procedure. The metrics measure the change in patient position by performing volume registration between every pair of temporally neighboring volumes in the rs-fMRI sequence. For each pair of subsequent volumes, one volume is designated as the stationary reference volume and the other becomes the moving volume. The moving volume undergoes a series of image transformations to find the best alignment of its contents to the reference volume’s contents. At least six transformation parameters must be used: three translation parameters and three rotation parameters. When the best alignment has been determined, the transformation parameters used to reach that alignment can be used to calculate, for example, the framewise displacement (FD). Three FD metrics calculated in this manner have been proposed by Power et al., Jenkinson et al., and Dosenbach et al. [17,18,19]. Comparisons of their metrics have shown that the FD metric suggested by Power et al. is approximately twice as large as the FD metric suggested by Jenkinson et al., and there is a high correlation between the FD metrics of Power et al. and Dosenbach et al. [19,20].

The change in the recorded signal is more difficult to quantify than the positional changes between image volumes. The magnitude of the recorded signal is impacted by both the spin history and susceptibility effects of motion, and it is difficult to separate the effects of these two causes. Generally, the signal difference between a pair of image volumes is calculated using the root mean square difference between the corresponding voxel values in both images. The framework used to choose the pairs of image volumes varies: signal intensity changes can be calculated using a single reference volume to compare to all other volumes in the image sequence or using temporally neighboring image volumes. The specific use of temporally neighboring image volumes for comparing signal changes was suggested by Smyser et al. [21]. Their metric measures the temporal derivative of the root mean squared variance over the voxel values (DVARS) between a pair of image volumes.

Generally, a “good” rs-fMRI sequence involves as little displacement of the brain as possible and does not involve true signal changes in each voxel greater than a small percentage of the global signal. The quantity of motion considered acceptable in an rs-fMRI sequence continues to be debated and guidelines for usability are expected to vary across developmental periods and scanning systems. Power et al. used their FD and Smyser et al.’s DVARS metrics to establish general usability guidelines for an image sequence: at least 50% of the image volumes must have both an FD of no more than 0.2 mm and a DVARS of no more than 2.5% signal intensity units from the previous volume [16]. Others have considered the total duration of low motion time present in the image sequence with recommended minimum timespans of low motion ranging from five to ten minutes across the sequence [22,23,24]. Regardless of its definition, low motion rs-fMRI scans can be difficult to obtain for pediatric patient populations, but effectively preventing patient motion or mitigating the effects of motion post-acquisition could increase the clinical applications of this modality.

## 5. Motion Prevention for MRIs

When a patient undergoes an MRI scan, a radiology technician will position them in the scanner so that the organ being scanned is stable. Thick foam pads of various shapes and sizes are often used to prevent the organ or body part from moving, though they do not completely immobilize the patient. The patient is instructed to remain still during the scan, but is still able to move. Several approaches have been developed to encourage a patient to remain still during a scan, though not all approaches will work for all patient populations.

### 5.1. Sedation

Though sedation is the only technique that can prevent all macro-level motions, studies have shown that sedation should not be used for rs-fMRIs. Sedation results in reduced brain activity and lower levels of consciousness. Stamatakis et al. used rs-fMRIs to collect BOLD data from healthy volunteers at three different levels of sedation: none, light, and moderate [25]. Their findings suggest brain activity during sedation mirrors activity observed in non-REM sleep, not activity observed, aware and task-free brains. Liu et al. observed reduced low-frequency fluctuations in BOLD signals in memory-related regions during light sedation and in the prefrontal cortex in deep sedation [26].

Furthermore, use of sedation during pediatric MRI scans is discouraged. Sedation is not recommended for use with neonatal and infant populations according to the Food and Drug Administration, and strict guidelines for pediatric sedation have been developed by The American Academy of Pediatrics and the American Academy of Pediatric Dentistry [27,28,29]. Generally, it increases the risks to the patient, does not change the amount of time spent in the scanner, and increases the amount of time the patient must spend in the scanning facility [30].

On the rare cases when sedation must be used to help a patient tolerate a scan, additional measures must be taken to ensure the patient’s safety. The patient may be instructed to limit their food and liquid intake to reduce the risk of pulmonary aspiration. Extra trained personnel must be present to administer the sedation and monitor the patient and MR compatible equipment must be easily accessible in case of emergent adverse reactions such as respiratory events, seizures, vomiting, and allergic reactions [28,29,31,32]. There must also be a place for the patient to recover from the sedation after the procedure where they can be monitored to ensure they recover to acceptable levels of consciousness and respiration. Once recovered, they must have safe transport with at least one responsible adult both to and from the hospital, and two adults are strongly recommended if the patient is transported using a car seat. These considerations all impact the cost per patient, time the patient and their family must spend at the facility, and the number of patients who can be scanned using a single scanner in one day.

Even though sedation may be the obvious solution for reducing patient movement, it can introduce bias into the BOLD signals recorded during an rs-fMRI due to altered brain activity. It is also not recommended for pediatric patients, especially infants and neonates. If sedation is the only option, it is costly: it increases the monetary and total time costs associated with a single patient’s scans, increases the risks a patient is exposed to during a scan, and does not significantly impact the amount of time the patient spends in the MRI scanner.

### 5.2. Patient Education, Training, and Distraction

Unless a patient has familiarity with the medical imaging field or has had a similar scan, it is unlikely that they will know what to expect when they attend their MRI scan appointment. Ideally, the patient and their family are provided with educational material to prepare them for the MRI experience. The success of patient education depends on how the material is presented to the patient as well as how the patient receives it. For pediatric patients or patients who are not neurotypical, reading a pamphlet may not be the best way to prepare the patient for the scan. In a review of literature focused on pediatric radiology procedures, Alexander describes several alternative techniques commonly used to prepare pediatric patients for different radiology procedures [33]. Replacing pamphlets with coloring books and short videos effectively exposes pediatric patients to the types of equipment they can expect to interact with during a procedure in a more accessible format.

MRI scanner simulators are another helpful tool for preparing a patient for an MRI scan. A patient may have the opportunity to practice undergoing an MRI scan upon arrival at the scanning facility. Realistic MRI simulators exist such as the simulator made by Psychology Software Tools (Psychology Software Tools, Sharpsburg, PA, USA), though Legos (The Lego Group, Billund, Denmark) and the Playful MRI Simulator (DOmed Medical Engineering, Lyon, France) may be used to help pediatric patients prepare for an MRI scan. Most simulators expose the patient to the physical appearance of the MRI scanner and help the patient practice remaining still in a small, enclosed space similar to how they would during a real MRI scan. However, the physical appearance and small space are only two of the aspects of an MRI scanner that may distress the patient; the sounds emitted by the MRI scanner can also be distressing. They vary from soft clicks to loud bangs depending on the acquisition protocol being used. Teams at the University of Michigan and the United Kingdom’s National Health Service have been working to develop virtual reality MRI simulators to offer a more comprehensive simulated MRI experience. They allow the patient to explore the MRI scanner, to practice laying in a virtual scanner while listening to acquisition-related sounds, and to learn more about the scan experience [34,35]. MRI simulators have been shown to be useful in helping patients prepare for their MRI scans [36].

Parents can help pediatric patients prepare for a scan in a variety of ways: practicing lying still, teaching behavioral coping techniques, and verbal reassurance during the scan. However, distress in a patient’s parent may increase the patient’s own distress [37,38]. Several researchers have found that interventions targeted toward reducing patient anxiety may also help reduce parental anxiety. Johnson et al. report that presenting educational coloring books to pediatric patients whose parents felt high levels of anxiety about the patient’s upcoming scan helped both the patients and the parents feel less anxious [39]. This study reported low levels of patient anxiety overall and suggests that the kid-friendly nature of the hospital environment may have biased the anxiety levels of the patients.

During a scan, distraction techniques can be used to calm and comfort the patient. Since as early as 1996, monitors displaying cartoons and videos have been used to help patients tolerate daily radiotherapy [40]. Moving light shows can be relaxing and distracting for younger patients [41]. More recently, headphones with music or stories and MR compatible video goggles have helped distract and comfort patients during their scans [33,36,42]. As personal entertainment technology evolves, some groups have begun investigating the potential role of virtual reality tools as distraction techniques for radiology. In a case study comparing the efficacy of music and immersive virtual reality tools as distractions during a mock scan, Garcia-Palacios et al. found that immersive virtual reality systems decrease patient anxiety during a scan more effectively than music alone [43]. As virtual reality technology improves, it may join headphones and MR compatible video goggles as an MR compatible distraction method. It should be noted that visual and auditory distractions may impact the activated regions in the brain, confounding the resting-state functional connectivity analysis [44].

Often, several of these techniques can be combined to create a more in-depth protocol for preparing a pediatric patient for an MRI scan. Here, we delve into further detail of four such studies.

Khan et al. evaluated sedation rates in pediatric CT and MRI scans before and after a sedation reduction program was implemented at the Cincinnati Children’s Hospital Medical Center [41]. Six components were introduced sequentially over a 12 month period, though only three of them were MR compatible. The MR compatible components were having a certified child-life specialist prepare, coach, distract, and support children during MRI scans, giving the patient MR compatible video goggles for watching and listening to movies during MRI scans, and implementing a clear departmental goal of reducing sedation. The combination of the components of the sedation reduction program decreased the overall rate of sedation for MRI scans in patients under the age of 7 from 80.8% to 52.8%. Patients who had experienced CT or MRI scans both before and after the sedation reduction program was implemented reported better experiences as a result of the implemented program.

Raschle et al. describe a protocol used to educate and train preschool aged patients for structural and functional MRI scans [45]. They emphasized three values (comfort, appropriateness, and motivation) to use when working with children. The protocol consists of several steps designed to educate and train the patient on what they should expect during the MRI scan. The 60 min training session was designed to be playful and engaging for the patient, and patient concerns were addressed as they arose during the training. After the training session, the patient was scanned for between 45 and 60 min. The authors found that dividing a long scan session into smaller acquisitions coupled with breaks resulted in more successful acquisition than long sessions with no breaks. Rewards such as stickers on a sticker chart, pictures of their own brain to take home, and a prize for completing the scan provided additional motivation for young patients. Over 95% of the patients who experienced this protocol were able to complete the scans without the need for sedation.

Klosky et al. evaluated two educational and distraction-based protocols for a study regarding parental and pediatric patient distress during radiotherapy procedures. They compared educational and distraction techniques that utilized the popular Barney the Dinosaur character in every step to similar techniques with no character continuity. They found that both parent anxiety and patient distress were decreased more in the educational and distraction experiences featuring Barney than in the generic experiences from the control group [46]. The consistent presence of Barney during the training and procedure, especially when the patient had to be alone in the room, provided comfort to the patient when the parent could not. However, the authors report in a related study that there was no significant change in the patient’s ability to endure the procedure without sedation [47].

Most recently, Horien et al. evaluated the efficacy of educational and “in-scanner” behavioral techniques with three groups of research patients. Patients were assigned to one of the following three groups depending on when they enrolled in the study: a control group with no intervention, a group receiving the intervention and scan from a single researcher, and a second intervention group where the intervention and scan were conducted by other members of the research team. This third group was included in the study design to measure the reproducibility of the intervention when conducted by other individuals. The intervention itself consisted of formal training with a mock scanner, the use of an MR compatible weighted blanket for children, and a flexible prize system for low motion scans. The findings showed that both intervention groups had on average lower frame-to-frame displacement than the control group, and the results of the intervention groups were robust to the individual administering the intervention [48].

A variety of behavior- and communication-based approaches have been developed to prevent patient motion. Each approach may be used on its own or combined with other approaches as needed. The success of the approaches described in this section is highly dependent on the age and maturity of the patient: successful scanning of an older toddler usually requires a larger degree of intervention than scanning of school-aged patients but each intervention should be customized to the patient’s individual needs.

### 5.3. Feed and Sleep Protocols

Though there is great potential for rs-fMRI research in neonatal patients, neonates can be difficult to scan as they move often and the suggestions in the previous section are not applicable to this age group: they are too young to understand or comply with instructions [49]. The one activity in which they remain relatively still is sleep. Aptly named, “feed and sleep” or “feed and bundle” protocols have been developed to take advantage of the stationary state of the sleeping neonate. Upon arrival at the scanning facility, the neonate and their parent(s) are shown to a private room where the neonate is fed. Shortly afterward, they are swaddled and, in most cases, fall asleep. They are then given hearing protection, immobilized, and placed in the scanner. The intention of this protocol is that the neonate will feel relaxed and comforted by the combination of the food and the swaddle and will sleep through the duration of the MRI scan.

Several variations of the feed and sleep protocol exist. One aspect of the protocol that may be different at different sites is when the neonate was last fed prior to their scan appointment. Some protocols call for the neonate to be deprived of food for a certain period leading up to the scan while other protocols time the scan appointment to align with the neonate’s feeding schedule. Windram et al. recommend a deprivation period of four hours while Gale et al. and Mathur et al. recommend timing the patient’s scan so that they will arrive at the scanning facility and be fed less than 45 min before their scan [50,51,52].

The materials used to secure the patient also vary. Windram et al.’s protocol uses a blanket swaddle and a vacuum-bag immobilizer. Gale et al.’s protocol emphasizes hearing protection by using dental putty, headphones, a hat, and foam padding to provide noise protection. Mathur et al.’s protocol calls for both strong ear protection and a vacuum-bag immobilizer.

### 5.4. Summary

The only technique which completely prevents a patient from actively moving is sedation. Sedation is often not an option because it adds risk to the procedure and additional burden on the patients, their family members, and the scanning facility. Safer alternatives to sedation have been developed, though they are not as effective in preventing patient movement.

Educating pediatric patients about what they can expect during an MRI scan can help prepare them for their scans. Patient education is often done using informational pamphlets, short videos, and coloring books for pediatric patients, and physical or virtual MRI scanner simulators. Different methods of patient distraction may be employed during the scan to distract the patient from distress or boredom. Training protocols using combinations of education, distraction, and comforting techniques have been found to be helpful in getting pediatric patients to tolerate an MRI scan, though they may require additional time commitments. For neonatal patients, a set of protocols involving feeding, swaddling, and immobilizing the patient have been developed.

## 6. Prospective Motion Correction

### 6.1. rs-fMRI Specific Approaches

Even if the protocols outlined in the previous section are used to prepare the patient in order to prevent motion correction, they do not actually prevent the patient from moving. The techniques which have been developed to monitor and correct motion during an MRI scan can be divided into three categories: optical motion correction techniques, correction using non-visual external sensors, and image signal based motion correction.

#### 6.1.1. Optical Motion Correction

Optical motion correction techniques use a combination of visually descriptive external markers and camera systems to monitor patient position during the MRI scan. One of the earliest examples of optical motion correction used a pair of cameras located outside the MRI scanner and a set of four reflective markers attached to a modified mouthpiece which the patient bit down on for the duration of the scan [53]. Changes in the position of the markers were processed during the scan and used to update the acquisition parameters in real-time. This setup was simplified once MR compatible cameras were developed: the two cameras outside the scanner were replaced with a single camera in the scanner bore. The reflective markers were then replaced with two-dimensional markers attached to the patient’s forehead. The 2D markers initially were plain chess board patterns, though they later were modified so that the computer could differentiate between the different blocks in the pattern [54,55]. Additionally, MR-detectible agar has been added to the chessboard so that the orientation of the marker could be detected in the signal acquired by the MRI scanner.

Now, development of optical motion correction techniques has shifted from research settings to industry settings. Several companies have developed computer vision-based solutions for optical motion correction. One company uses a high-resolution MR-compatible camera and a marker attached to the patient’s nose to detect motion and monitor respiration (KinetiCor Biometric Intelligence, Honolulu, HI, USA). Another company uses a stereo camera system to record a point cloud of the features of a patient’s face and generate a primary marker. Small facial movements as well as cardiac and respiratory motion are monitored using the point cloud while larger movements are monitored using both the point cloud and the primary marker (TracInnovations, Ballerup, Denmark).

#### 6.1.2. External Sensors

Other external sensors can be used to monitor different aspects of patient movement. Wired magnetic resonance field probes, wireless inductivity-coupled markers, and off-resonance markers directly interact with the magnetic field of the MR scanner. One study used a tracker consisting of two sensors attached to the patient’s forehead (Robin Medical Inc., Baltimore, MD, USA) to measure the position and orientation of the patient relative to the center of the scanner bore [56]. Depending on the sensor, the information recorded by it may be available during or after the scan.

In some cases, it may be useful to record information about a patient’s physiological motion during the scan. Respiratory bellows can be used to monitor the patient’s respiration activity and potentially instruct the computer to only perform acquisitions during a certain stage of the respiratory cycle. If both respiration and cardiac activity need to be recorded, a combination of the bellows or a pressure belt around their chest and a pulse oximeter on their finger can be used to record these types of activity [57]. The information recorded by these sensors is available after the scan to aid in correcting for physiological motion.

#### 6.1.3. Within MRI Motion Correction

Intra-image motion correction is an area of prospective motion correction focused on updating pulse sequences during image acquisition. Early research on intra-image motion correction dates back over thirty years. A detailed timeline of early developmental milestones can be found in Maclaren et al.’s review of prospective motion correction in brain imaging [58]. One notable highlight is the work done by Thesen et al. They designed a technique called PACE (Prospective Acquisition Corr Ection) which used real time estimations of rotation and translation to adjust slice position and orientation between the acquisition of individual fMRI volumes [59]. This technique was developed into a commercial product, Siemens PACE.

Since Maclaren et al.’s review, one group has developed a tool to evaluate motion in rs-fMRI sequences as they are acquired using the motion metric FD discussed above [19,60]. Their tool performs volume registration between the first volume of the rs-fMRI sequence and the most recently acquired image volume. The parameters calculated via volume registration are used to calculate the FD between the current volume and the first volume. The calculated FD is then compared to a set of displacement thresholds associated with the scan’s quality. The FD is displayed for the radiology technician and is used to either ensure the scan contains a usable quantity of low motion volumes or to terminate the scan early if obtaining a usable quantity of low motion volumes is not possible. This information can also be used to adjust the acquisition protocol parameters. Though the process of integrating the tool with a scanner’s software can be difficult, it has the potential to reduce scan time and costs by at least 50% [19].

The methods and technologies discussed in this section have a few limitations. The optical motion correction methods are limited by the physical set up and MR compatibility of the cameras and the markers. Sequences may need to be modified to account for non-visual external sensors, which can complicate the MRI scan process. Additionally, some prospective motion correction techniques have the potential to corrupt the scan, and it is recommended that the prospectively corrected scan is acquired independently from a second, correction-free scan [61].

### 6.2. Non rs-fMRI Approaches

Though this review focuses on motion in rs-fMRIs, we include a brief overview of techniques used in non-fMRIs. As the field continues to grow and new methods to solve the problem of motion emerge, some of these techniques may find new applications in the fMRI domain.

#### 6.2.1. Optical Motion Correction

The techniques discussed above can also apply to non-rs-fMRI acquisitions.

#### 6.2.2. External Sensors

One of the most common techniques for mitigating the effects of physiological noise in structural MRIs is gating. Gating uses physiological activity to produce an image in which all data was recorded when the patient was in the same physiological state. Gating techniques measure cardiac and/or respiratory activity during the scan. This information is used either during the scan to adjust acquisition timing or after the scan to filter acquired data so that the resulting sequence contains only data from the same points in the cardiac and/or respiratory cycles.

Gating techniques are not applicable to functional imaging as they introduce spin history effects on the temporal scale of the BOLD signal. However, research in the area of modeling cardiac and respiratory noise has the potential to eliminate the need for gating techniques. Early work by Biswal et al. uses a pulse oximeter to record cardiac and respiratory information during fMRI acquisition. The sensor information was used to generate a temporal passband filter to remove data from cardiac and respiratory “off cycles” from the fMRI sequence [62]. More recently, Liston et al. suggest that cardiac noise can be estimated and modeled using ECGs. Removal of this noise has the potential to improve the truth of the signals in cardiac affected voxels by 18.5 ± 4.8% [63].

#### 6.2.3. Within MRI Motion Correction

Structural MRI scans are also sensitive to motion, but there are more prospective motion correction techniques that can be used in structural imaging due to the large amount of “dead time” in many structural sequences to allow for appropriate tissue contrast to evolve.

One such technique is the navigator sub-sequence. A navigator sub-sequence is a quick, low-resolution image that can be used to determine the position and orientation of the patient’s head in the scanner bore. The parameters of the acquisition protocol are then modified on-the-fly to maintain the alignment of the imaging axes with the patient’s head [64,65,66,67]. Registering the most recent navigator image to the previous navigator produces a measure of the patient’s motion. An analysis of this navigator-based approach to prospective motion correction suggests that it significantly reduces the effects of motion in structural images and diffusion tensor images [68,69].

Though the field of rs-fMRI acquisition is evolving, current navigator sequences increase the relaxation time of an MR protocol too much to inject them into rs-fMRI sequence acquisition. The integration of multiband acquisition techniques into rs-fMRI protocols may provide the opportunity for navigator-based prospective motion correction to be translated into the functional domain.

## 7. Retrospective Motion Correction for rs-fMRIs

Many groups have put significant effort into developing techniques for recovering motion corrupted images. Here, we discuss several retrospective motion correction techniques in the general categories of volumetric image registration, denoising, and filtering.

Volumetric image registration is the first step in any retrospective motion correction pipeline. As discussed earlier in Section 4, the registration process begins with identifying a stationary reference volume in the rs-fMRI sequence. Every other volume in the sequence then undergoes various spatial transformations so that their contents align optimally with the contents of the reference volume. Different researchers have used different volumes from the image sequence as the reference volume. Friston et al. designated the first volume in the image sequence as the reference volume, though other common choices include the average of all volumes in the sequence, the middle volume in the sequence, and the volume most similar to the rest of the sequence [70,71]. The transformations which can be applied to the moving volume include translation, rotation, scaling, skewing, and nonlinear adjustments. The linear and affine transformations reduce large scale differences between image volumes while the nonlinear transformations can fine-tune differences in smaller neighborhoods of voxels. It should be noted that as volume registration is an optimization process, it is subject to problems inherent to any optimization process. The choice of optimization algorithm and cost function may impact the results of the registration.

Variations on the traditional “all to one” registration method have been developed. Recently, Liao et al. suggested a registration framework based on the concept of a hidden Markov model [72]. In their framework, the information about the previous volume’s registration was used to initialize the registration of the current volume.

After an image sequence has undergone volume registration, it can undergo denoising. Denoising consists of identifying correlations between BOLD signal components in the rs-fMRI sequence and other factors and then removing them from the rs-fMRI using regression. One popular set of regression factors are the translation and rotation parameters obtained during volume registration and their first order derivatives [17,22,73]. Variations of this set of factors have also been used, including the transformation parameters and derivatives of the previous or subsequent volumes [16,20,74]. Another popular set of regression factors are signals identified using techniques such as principal component analysis or independent component analysis, which decompose the signals present in the rs-fMRI into lists of components [75,76,77,78]. A third group of regression factors include signals associated with physiological properties of the sequence. These properties include the global signal, white matter signal, and cerebrospinal fluid signal [16,20,73,74,79,80]. There is a tradeoff when removing global signals: removing white matter and cerebrospinal fluid signals reduces the effects of motion on the BOLD signal but removing the global signal may weaken shorter distance neuronal network connections [81,82]. However, techniques such as CompCor that remove signals from areas of regions of no interest show improved motion mitigation [78,83].

After registration and denoising, filtering techniques may be used to further reduce the effects of motion. Filtering techniques detect and remove motion-related signal outliers from the sequence. Three categories of filtering techniques are scrubbing, spike regression, and despiking.

Scrubbing first examines the sequence for image volumes with high FD, then for each high motion volume removes the high motion volume, the previous volume, and the following pair of volumes [17]. The remaining subsequences can potentially be concatenated without negatively impacting later analyses of the BOLD signal if the shortened sequence contains at least 125 frames [22,84]. Many variations on the idea of temporal filtering have been developed, though all of them result in shorter rs-fMRI sequences [21,85,86,87,88,89].

Spike regression identifies image volumes with FD or FD and DVARs metrics which surpass a given threshold or thresholds [74]. The thresholds must be chosen with care: lower thresholds will identify more signal spikes and remove more data while higher thresholds will retain more data and identify fewer spikes. The spikes are modeled as signals to regress out of the image sequence, similar to the denoising techniques discussed previously.

While the first two filtering techniques viewed the rs-fMRI sequence as a list of image volumes, despiking treats the rs-fMRI as a single volume where each voxel contains a temporal signal. Each voxel’s temporal signal is examined for sudden value changes. These spikes are replaced using interpolated values calculated using the time points in the neighborhood around the spike [80,90]. Despiking does not remove whole volumes as scrubbing and spike regression do, but it is more sensitive to the effects of motion on a voxel level.

In addition, some dynamic field distortion correction methods to compensate for spin history and susceptibility effects have been studied in a few specific cases, but their impact has yet to be studied on a broader scope [61]. Combining solutions to these effects of motion with the methods discussed in this section has the potential to retrospectively solve the problem of motion in rs-fMRIs.

## 8. Discussion

The purpose of this review is to explore the breadth of the technical and clinical challenges of performing rs-fMRI scans on pediatric patients and present them in a comprehensive and approachable report. Other reviews of motion in pediatric MRI have been performed, though they usually go into more detail on a narrower focus. Wilke et al. identified several challenges and proposed solutions associated with functional and diffusion pediatric MR scans [91]. Maknojia et al. suggested that the best stage of the MR acquisition process to address motion is after scan acquisition, though the combination of prospective and retrospective correction methods show promise [92]. An entire book (Handbook of Pediatric Brain Imaging: Methods and Applications) has been written on the topic [93].

Resting-state fMRI sequences have the potential to aid in the diagnosis of neurodevelopmental disorders, though their current clinical use is limited. These sequences record the fluctuations in the BOLD signal in the brain, which approximates brain activity, over a period of time. The BOLD signal recorded by the MRI scanner is highly sensitive to patient motion. The effects of motion fall into three categories: positional effects, spin history effects, and susceptibility effects. These effects can be measured using the FD and DVARS metrics.

Patient motion can be prevented using sedation, but sedation is not recommended for pediatric patients and can introduce sedation-specific bias into the signal. Combinations of educational training and distraction techniques have been shown to be helpful in helping pediatric patients tolerate MRI scans with less motion. Neonatal patients can be scanned while asleep after feeding, which reduces the likelihood that they will move. Even with these protocols, prospective and retrospective motion correction techniques have needed to be developed to reduce the positional and signal related effects of motion both during and after image acquisition. Ultimately, the amount of motion present in the image sequence after acquisition and cleaning determines whether or not it can be used in clinical or research applications.

## 9. Conclusions

Patient motion is a problem which affects every stage of rs-fMRI acquisition. However, most of the research done regarding this problem falls into silos depending on where in the acquisition process the researchers plan to address the motion. Research in this area is usually divided into sets of behavioral and educational scanning protocols, prospective motion monitoring and correction, and retrospective motion correction. The true challenge of motion in pediatric rs-fMRI is not that we are not doing enough to address it, but rather that we do not have solutions built upon a robust understanding of the various facets of the problem. Each discussion with a physicist, clinician, study coordinator, and data scientist revealed new nuances related to motion in pediatric rs-fMRI studies. As such, further collaborative work is needed to mitigate the effects of motion in pediatric rs-fMRI.

## Figures and Tables

**Figure 1 diagnostics-12-01032-f001:**
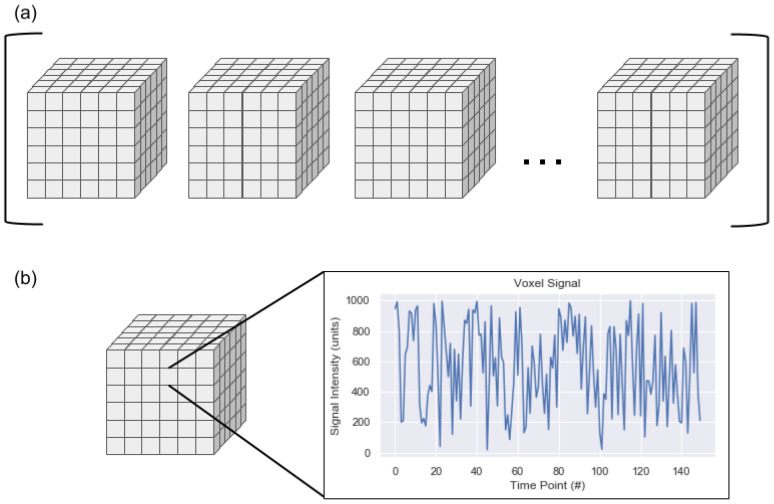
An rs-fMRI can be thought of as (**a**) an ordered list of three-dimensional image volumes where each voxel has a decimal value or as (**b**) a single three-dimensional image volume where each voxel contains a temporal signal.

**Figure 2 diagnostics-12-01032-f002:**
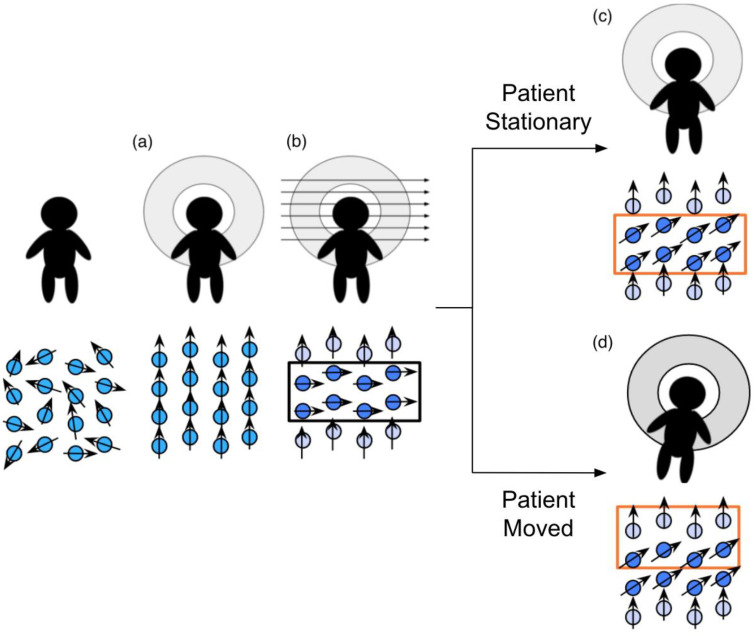
The process by which patient movement causes the spin history effect. When the patient enters the scanner, (**a**) all molecules with any polarity align to the magnetic field. (**b**) An RF excitation pulse is applied to a limited volumetric space within the patient, forcing those molecules to align with the temporary, secondary magnetic field. When the patient is stationary (**c**), the MR scanner records EM signals from those excited molecules. If the patient moves (**d**), the MR scanner records EM signals from a mix of intentionally excited and previously excited molecules.

## Data Availability

Not applicable.

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
