# Peer review of "A Descriptive Review of the Impact of Patient Motion in Early Childhood Resting-State Functional Magnetic Resonance Imaging"

_diagnostics, 2022, doi:10.3390/diagnostics12051032_

Round 1
Reviewer 1 Report
Comments are attached.

Author Response
This review addresses the very timely and important topic of the effects of motion in pediatric resting state functional magnetic resonance imaging. The authors have done a commendable job of covering multiple aspects of a very complex topic, including MR signal generation, different types of motion artifacts, measurement of motion, protocols to reduce motion, and prospective and retrospective motion correction. The manuscript could be improved through clarifying the overall purpose of the review, including the intended audience, and addressing related issues with organization.
- What is the overarching purpose of the review? The authors state: “This work is intended as a descriptive overview of rs-fMRIs, how they are impacted by motion, and methods used to prevent and correct motion in pediatric MR imaging.” However, as they note themselves in the Discussion, there are multiple articles and books that have tackled this topic already. It will be important to clarify what this review adds to the literature.
Response: The reviewer brings up a good point – other researchers in the field have addressed the problem of motion in rs-fMRIs. This review article is intended to provide a scientifically sound yet understandable overview of the breadth of the problem while providing suggestions of articles and books that provide more depth in specific areas of the problem. We have expanded the description of the purpose of this review in lines 59-65 and restructured the first paragraph of section 8 (Discussion) to more clearly articulate the intention of this review.
- Related to the prior point, who is the intended audience of this review? It seems to be somewhat ambiguous as the authors refer to the potential clinical utility of fMRI, but also state that it is not actually used clinically.
Response: Please see the suggested edits in lines 59-65.
- The description of task-based and rs-fsMRI is very cursory and appears to come as an afterthought (lines 119-125). I would recommend including a fuller description of rs-fMRI and what it has revealed about brain organization. This would better set the stage for why it is important to consider how motion artifacts may impact rs-fMRI. At present it is not clear why the review focuses on rs-fMRI. I also think it would flow better to move the description of non-motions artifacts to after the description of rs-fMRI. As written, rs-fMRI appears to be an afterthought following the description of artifacts.
Response: Thank you for this suggestion. Two paragraphs have been added to the beginning of section 2 (lines 73-92) to elaborate on the role rs-fMRI has played in the study of the default mode network and functional connectivity in various diseases and disorders.
- The discussion of the amount of low motion data required for rs-fsMRI is superficial to the extent that it is not useful. It should either be expanded to explain the rationale behind different cutoffs for the amount of low motion data and the cost/benefit of different scan lengths, or removed.
Response: Based on the reviewer’s suggestion, we have reduced the discussion of the duration of low motion data and added a smoother transition to the following section (lines 285-291).
- It seems odd to have the first section on motion prevention focus on sedation considering that it is not used in research with pediatric populations and increasingly discouraged in clinical settings. The general focus on sedation seems quite outdated.
Response: Though the use of sedation is not encouraged, we would be remiss if we did not include a brief discussion of it, why it should not be used in pediatric rs-fMRIs, and the precautions that must be considered in cases where there is no other option. The intended audience of this review may not be familiar with these issues. Section 5.1 has been edited accordingly (please see lines 301-361) and the names of subsequent sections in 5 have been updated.
- It is difficult to lump all pediatric patients together in discussing approaches to motion prevention given that the approaches are so different even between neonates versus older infants and toddlers, and then completely different in school aged children and older.
Response: We try to address this challenge by separating behavior based techniques from the feed and sleep techniques and have added lines 621-626 to clarify the potential applications of the behavioral techniques.
- Overall, the review would benefit from being more focused and organized. It is currently unclear why all of the different topics and sections are necessary as they range from the basics of MRI and fMRI signal generation to protocols for prevention of movement across different pediatric age ranges to motion correction strategies for different imaging modalities.
Response: Transitions between sections have been rewritten to smooth the transitions between topics. With regards to the breadth of this review and its intended audience, the variety of topics is expected.

Reviewer 2 Report
Very complete and comprehensive review.
The problematic of the impact of the patient motion in early childhood resting state functional magnetic resonance imaging wa revised in every its parts, even the smallest one, and from all the points of view.
It helps to fully understand the problem and therefore can receive positive feedback from the scientific community.
Author Response
Thank you for your comments.